# New Mitogenomes of the *Polypedilum* Generic Complex (Diptera: Chironomidae): Characterization and Phylogenetic Implications

**DOI:** 10.3390/insects14030238

**Published:** 2023-02-27

**Authors:** Dan Zhang, Fei-Xiang He, Xue-Bo Li, Zhulidezi Aishan, Xiao-Long Lin

**Affiliations:** 1Characteristic Laboratory of Forensic Science in Universities of Shandong Province, Shandong University of Political Science and Law, Jinan 250014, China; 2Dongting Lake Eco-Environmental Monitoring Center of Hunan Province, Yueyang 414000, China; 3College of Life Science and Technology, Xinjiang University, Urumqi 830017, China; 4Engineering Research Center of Environmental DNA and Ecological Water Health Assessment, Shanghai Ocean University, Shanghai 201306, China; 5Shanghai Universities Key Laboratory of Marine Animal Taxonomy and Evolution, Shanghai Ocean University, Shanghai 201306, China

**Keywords:** Chironomidae, mitogenome, *Polypedilum*, phylogenomics

## Abstract

**Simple Summary:**

Many species from the *Polypedilum* generic complex are renowned for their roles in aquatic ecosystems. This genus complex, as one of the most species-rich groups, has been persistently contentious. The current lack of sequence data for the *Polypedilum* complex limits our comprehension of their species evolution. Here, 14 mitogenomes of the *Polypedilum* generic complex were sequenced. Combined with three recently published mitochondrial genomes, we analyzed the features of the mitogenomes among genera. Furthermore, the phylogenetic analysis among genera within the *Polypedilum* complex showed that the *Endochironomus* + *Synendotendipes* is the sister group of *Phaenopsectra* + *Sergentia*. Our study provides a vital foundation for further study on the evolutionary biology of Chironomidae.

**Abstract:**

Mitochondrial genomics, as a useful marker for phylogenetics and systematics of organisms, are important for molecular biology studies. The phylogenetic relationships of the *Polypedilum* generic complex remains controversial, due to lack taxonomy and molecular information. In this study, we newly sequenced mitogenomes of 14 species of the *Polypedilum* generic complex. Coupled with three recently published sequences, we analyzed the nucleotide composition, sequence length, and evolutionary rate of this generic complex. The control region showed the highest AT content. The evolution rate of protein coding genes was as follows: *ATP8* > *ND6* > *ND5* > *ND3* > *ND2* > *ND4L* > *ND4* > *COX1* > *ND1* > *CYTB* > *APT6* > *COX2* > *COX3*. We reconstructed the phylogenetic relationships among the genera within the *Polypedilum* generic complex based on 19 mitochondrial genomes (seventeen ingroups and two outgroups), using Bayesian Inference (BI) and Maximum Likelihood (ML) methods for all databases. Phylogenetic analysis of 19 mitochondrial genomes demonstrated that the *Endochironomus* + *Synendotendipes* was sister to *Phaenopsectra* + *Sergentia*.

## 1. Introduction

Mitochondrial genomes are important molecular markers that have usually been used for studies on phylogeny, evolutionary history, speciation and phylogeography in insect groups [1,2,3], benefit by the maternal inheritance, high substitution, and easy availability [4,5]. Mitogenomes are known as the second genetic information system, because of its special genetic characteristics [6]. The mitochondrial genome length of insects ranged from 14,000 to 20,000 bp mostly [1], containing two ribosomal RNAs (rRNAs), thirteen protein-coding genes (PCGs), 22 transfer RNA (tRNAs), and one non-coding control region (CR) [7]. The mitogenomes structure of insects are conserved, with the same genes arrange as the ancestral insect. In addition, the structural features of mitogenomes can provide more information and evidence for morphological classification. The number of complete mitogenomes of insects and Chironomidae have gradually increased, quickening with advances in next-generation sequencing, to resolve the evolutionary history and structure comparison in different taxonomic levels [3,5,7,8,9,10]. 

Chironomids is one of the most important aquatic insect groups. Almost all Chironomidae species have well-developed resilience and resistance in environmental stressors [3,5,11,12]. Chironomidae is a remarkable cosmopolitan and diverse group, with more than 7500 described species worldwide [11]. Their larvae inhabit varied habitats, including semi-aquatic, terrestrial, and freshwater habitats [12]. Most species sensitively perceive the environmental changes in trophic state temperature, and salinity or acidity, which make them valuable indicator organisms for the aquatic ecosystem [13]. 

The *Polypedilum* generic complex belongs to the tribe Chironomini of the subfamily Chironominae, containing *Ainurusurika*, *Endochironomus*, *Endotribelos*, *Phaenopsectra*, *Polypedilum*, *Sergentia*, *Stictochironomus*, *Synendotendipes*, *Tribelos*, and *Zhouomyia* [14,15]. Species of this group are mainly distributed in neotropical rivers [16]. They can tolerate polluted waters temperately, making them a biological indicator of environments [17]. Their diversity of larval habitat, which includes wood-mining, and aquatic habit therein, also render them an important group for studying evolutionary biology [12,17]. Several taxonomy and molecular studies have examined the phylogenetic relationships of the *Polypedilum* generic complex [15,18,19], but uncertainties in classification schemes among these groups still persist [19,20,21]. 

Recently, there were more studies conducted on mitochondrial genomes, which greatly promoted the research progress on systematics evolutionary history in Chironomidae. In addition, the increasing number of available mitogenomes of Chironomidae also assist to explore the Chironomidae mitochondrial structure and evolution pattern [3,5,10], and provide more taxonomic characters. Lin and his colleagues, for example, used mitochondrial genomes to reconstruct the phylogenetic relationships of Prodiamesinae (Diptera: Chironomidae); the results indicated that Prodiamesinae is a subgroup of Orthocladiinae [3]. However, mitogenomes resources of the *Polypedilum* generic complex are rare. To date, only a few mitogenomes in the *Polypedilum* generic complex were available, representing only the genera *Polypedilum* and *Stictochironomus* [22,23,24,25]. 

Herein, we newly sequenced, assembled, and annotated 14 species of six genera from the *Polypedilum* generic complex, and analyzed the characters of their mitogenome. Combined with three previously published mitogenomes, we compared the main features, substitution, and evolutionary rates of the mitogenomes among the *Polypedilum* generic complex. Finally, we reconstructed the phylogenetic relationships of this group based on 19 mitochondrial genomes (17 ingroups and two outgroups), using Maximum Likelihood (ML) and Bayesian Inference (BI) approaches.

## 2. Materials and Methods

### 2.1. Taxon Sampling and Sequencing

To reconstruct the phylogenetic relationships of the *Polypedilum* generic complex, our taxon sampling included most genera in this group. Here, our analysis included 17 taxa of the *Polypedilum* generic complex. We newly sequenced 14 species, including four *Polypedilum* species, three *Stictochironomus* species, three *Endochironomus* species, two *Synendotendipes* species, one *Phaenopsectra* species, and one *Sergentia* species, which were collected from China, Italy, and Norway by X.X.L. during 2013–2021 (detailed information shown in Table 1). In addition, mitogenomes of three *Polypedilum* species were retrieved from GenBank for comparative mitogenomic and phylogenetic analyses (Appendix A). Based on prior phylogenetic studies of Chironomidae [20], two species of the closely related genus *Stenochironomus* were selected as outgroups. In total, we sampled 19 species of six members of the *Polypedilum* generic group, including *Endochironomus*, *Phaenopsectra*, *Polypedilum*, *Sergentia*, *Stictochironomus*, and *Synendotendipes* (detailed information showed in Table 1 and Appendix A). All samples were immersed in 85% to 95% ethanol at −20 °C before DNA extraction and morphological examination. Specimen identifications were made by X.L.L. All vouchers were deposited in the College of Fisheries and Life, Shanghai Ocean University, Shanghai, China.

Whole genome DNA was extracted with Qiagen DNeasy Blood & Tissue Kit, and the concentration of the DNA was measured with a Qubit^®^ DNA Assay Kit in Qubit^®^ 2.0 Flurometer (ThermoFisher, USA). All whole genomes were sent to company for sequencing (Berry Genomics, Beijing, China). Truseq Nano DNA HT sample preparation Kit (Illumina, USA) was used to generate sequencing libraries. Raw reads were sequenced on the Illumina NovaSeq 6000 platform with 150 bp paired-end reads and were generated with an insert size around 350 bp. Adapters, short, and low-quality reads of raw data were removed using Trimmomatic v0.32 (Jülich, Germany) [26].

### 2.2. Assembly, Annotation and Composition Analyses

To ensure accuracy, we used two methods for de novo assembly: (1) NOVOPlasty v3.8.3 (Brussel, Belgium) [27] was implemented for mitogenome assembly with *COI* gene of *Polypedilum heberti* (GenBank accession: MK505566) as seed sequences and k-mer sizes of 23–39 bp; and (2) IDBA-UD v1.1.3 (Boston, MA, USA) [28] was used to assemble with parameter “--mink 40 --maxk 120”. Geneious 2020.2.1 [29] was used to compare mitogenome sequences which were obtained by the above two methods, and combine them into a single sequence. The secondary structure of tRNAs was implemented in tRNAscan SE 2.0 [30] and MITOS WebServer. The rRNAs and PCGs were annotated manually with the *Polypedilum vanderplanki* (GenBank accession: KT251040) as a reference using Clustal Omega in Geneious. Clustal W function in MEGA 7 was used to proofread the boundaries of rRNAs and PCGs [31]. Bias of the nucleotide composition and nucleotide composition of each gene was performed via SeqKit v0.16.0 (Chongqing, China) [32]. 

Rates of non-synonymous substitution rate (Ka)/synonymous substitution rate (Ks) for each PCG were calculated via DnaSP 6.0 [33]. Two formulas were used to calculate AT-skew and GC-skew: AT-skew = (A − T)/ (A + T), GC-skew = (G − C)/ (G + C). An online server CGview (https://cgview.ca/, accessed on 30 November 2022) was used to generate the mitogenome map to show sequence features. Finally, these 14 new mitogenome sequences were deposited in GenBank (for accession numbers, see Table 2).

### 2.3. Phylogenetic Analyses

A total of 2 rRNAs and 13 PCGs genes of 19 mitogenomes were used for phylogenetic analyses. Nucleotide and protein sequences were aligned via MAFFT v7.450 (Osaka, Japan) [34] with the L-INS-I method. Trimming was performed by Trimal v1.4.1 (Barcelona, Spain) [35] with “-automated1” strategy, and then we concatenated five matrices via FASconCAT-G v1.04 (Santa Cruz, CA, USA) [36] for phylogeny analysis: (1) cds_faa matrix contained all PCGs amino acid reads; (2) cds_fna matrix included all PCGs nucleotide reads; (3) cds_rrna matrix included all PCGs and two rRNA nucleotide reads; (4) cds12_fna matrix contained all PCGs nucleotide reads except the third codon positions; and; (5) cds12_rrna matrix contained PCGs nucleotide reads which removed the third codon positions and two rRNA gene. AliGROOVE v1.06 (Bonn, Germany) [37] was used to calculate the heterogeneity among matrices. 

For all matrices, we used ML and BI approaches to infer the phylogenetic relationships of the *Polypedilum* generic complex. For the ML analysis, we used ModelFinder [38] to select the best-fitting substitution models implemented in IQ-TREE 2 (Canberra, ACT, Australia) [39]. The posterior mean site frequency (PMSF) [40] model was used to minimize long-branch attraction artifacts for matrix cds_faa, with the command ‘−m − mtART + C60 + FO + R’ in IQ-TREE. Phylobayes-MPI (Montréal, QC, Canada) [41] was implemented to generate the BI tree, with the site-heterogeneous mixture model (−m CAT + GTR). We performed two Markov chain Monte Carlo chains (MCMC) with 10,000,000 generations, and stopped when we achieved satisfactory convergence (maxdiff < 0.3). A total of 25% initial trees of each run were discarded as burn-in, and we generated a consensus tree using the remaining trees combined. iTOL, an available online website, was used for tree beautified (https://itol.embl.de/upload.cgi, accessed on 15 December 2022). 

## 3. Results and Discussion

### 3.1. Mitogenomic Organization

We sequenced about three Gb raw reads for each sample. A total of 14 mitogenomes of Chironomidae were obtained in this study, of which five were complete mitogenomes and nine were linear mitogenomes, and all of them were submitted in GenBank with accession number OP950216–OP950228, OK513041 (Table 2 and Appendix A). The whole length of newly obtained sequences ranged from 15,582 (*Polypedilum masudai*) to 17,810 bp (*Sergentia baueri*), the variation in which was mainly caused by the unstable size of the CR (Table 2). All newly assembled mitogenomes contained the typeset of one CR and 37 genes which included 13 PCGs, 22 tRNAs, and two rRNAs (Figure 1). Most of the newly assembled mitogenomes were similar to those previously published for Chironomidae in length [3,21,22,23,24]. Sequence features of the represented species are given in Figure 1. 

The nucleotide composition of the newly reported mitogenomes were similar (Table 2), revealing the characteristic AT-biased composition in Chironomidae and other insects [2,3,10,25,42,43,44]. The AT content of the mitochondrial genomes ranged from 75.98% (*Stictochironomus akizukii*) to 79.94% (*Phaenopsectra flavipes*; Figure 2; Table 2). The CR showed the highest AT content, ranging from 91.95% (*S. akizukii*) to 98.18% (*Endochironomus albipennis*). The AT content in tRNA and PCGs was lower than that in rRNAs (Table 2). All newly assembled mitogenomes had a negative GC-skew, while the AT-skew for most of them was positive, except *Endochironomus tendens* which showed negative AT-skew. The GC-skew ranged from −32.02 (*S. akizukii*) to −16.30 (*Synendotendipes* sp.1), and the AT-skew ranged from −0.002 (*Polypedilum masudai*) to 0.075 (*S. akizukii*); the GC content (%) ranged from 20.06 (*P. flavipes*) to 24.02 (*S. akizukii*) (detailed information is given in Table 2). 

### 3.2. Protein-Coding Genes, Codon Usage, and Evolutionary Rates

There was no remarkable difference in the size of tRNA, PCGs, and rRNAs among each species. A total of 13 PCGs of obtained mitogenomes ranged from 11,196 (*Polypedilum unifascium*) to 11,241 bp (*Endochironomus pekanus*) in length. Combined and compared with published Chironomidae data, we found that the AT content of the third codon positions was significantly higher than the first and the second positions in PCGs (Figure 2). Most of the 14 mitogenomes exhibited negative GC-skew in PCGs, while *E. tendens* and *E. pekanus* were positive; and each of them had negative AT-skew of PCGs, ranging from −0.21 (*P. masudai*) to −0.17 (*E. tendens*). The AT content (%) ranged from 71.53 (*S. akizukii*) to 76.93 (*E. albipennis*); the GC content (%) ranged from 23.07 (*E. albipennis*) to 28.47 (*S. akizukii*) (for detailed information, see Table 2). 

All 13 PCGs of newly obtained mitogenomes had the standard start codon ATN, which was most similar to insect mitochondrial. However, several different start codons were found; the start codon of *COI* gene in 16 species was TTG, in two species was ATG, and in one species was ATA; *ATP8* gene in five species was ATA, in nine species was ATT, and in five species was ATC; *ND2* was ATT in all species, the *ND5* gene in 12 species was GTG, in six species was ATG, in one species was ATC and so on, and detailed statistical information as shown in Appendix A. The codon size ranged from 110 (*Endochironomus albipennis*) to 1180 bp (*Polypedilum heberti*) (Table 1). 

The Ka/Ks value (ω) was usually used to measure the rate of sequence evolution by natural selection [45,46]. The result of Ka/Ks ratio of all 13 PCGs was less than one, ranging from 0.0958 (*COX3*) to 0.7251 (*ATP8*) (Figure 3), which was same as other insects [43,44]. The evolution rate of 13 PCGs was as follows: *ATP8* > *ND6* > *ND5* > *ND3* > *ND2* > *ND4L* > *ND4* > *COX1* > *ND1* > *CYTB* > *APT6* > *COX2* > *COX3*. This result indicated that, in most cases, selection eliminated deleterious mutation, and all of them evolved under purifying selection pressure (Figure 3). In PCGs, each gene was under different purifying selection. *ATP8*, *ND6*, and *ND5* showed higher ω value, indicating that they exhibited relatively relaxed purifying selection pressure. Meanwhile, *COX2* and *COX3* were under the hardly purifying selection, which were similar to previous research results regarding chironomids [2,3,10,43].

An amount of 22 tRNAs ranged from 53 to 77 bp in length, the AT content (%) ranged from 79.17 (*S. akizukii*) to 82.04 (*Polypedilum vanderplanki*); except *Polypedilum unifascium* (−0.007), all others exhibited a positive AT-skew, ranging from 0.016 to 0.063; the GC content (%) ranged from 17.96 (*P. vanderplanki*) to 20.83 (*S. akizukii*); the GC-skew ranged from 11.54 (*S. akizukii*) to 17.36 (*Synendotendipes* sp.1). rRNA lengths ranged from 2224 in *Synendotendipes* sp.1 to 2342 bp in *S. akizukii*; the AT content (%) ranged from 13.81 to 15.68; the GC content (%) ranged from 11.30 to 15.68; the GC-skew of all mitogenomes were obviously positive (28.79 to 39.01); the AT-skew in most mitogenomes were negative (−0.048 to −0.002), while five species were negative (*P. heberti*, *Polypedilum yongsanensis*, *E. pekanus*, *S. akizukii*, and *P. nubifer*) for more detailed information, see Table 2). 

### 3.3. Phylogenetic Relationships

The heterogeneous divergence analysis indicated that the matrix cds12_rrna and cds_rrna exhibited higher heterogeneity than cds_faa, cds12_fna, and cds_fna (Figure 4). Because of high heterogeneity, third codon positions were rejected to reconstruct the phylogenetic relationship of the *Polypedilum* generic complex. Furthermore, the genera *Endochironomus* and *Synendotendipes* exhibited lower heterogeneity than *Polypedilum*, *Stictochironomus*, and *Sergentia* (Figure 4). 

We used supermatrix cds_faa (3715 sites), cds_fna (10,833 sites), cds_rrna (13,368 sites), cds12_fna (7430 sites), and cds12_rrna (9653 sites) to reconstruct phylogenetic relationships within the *Polypedilum* generic complex by two different methods. BI and ML approaches based on five matrices yielded five and six trees, respectively (Figure 5 and Appendix A), representing three different topologies. Our result was consistent with phylogenetic tree inferred from previous studies based on four genetic markers [20]. The monophyly of the genera *Endochironomus*, *Polypedilum*, *Stictochironomus*, and *Synendotendipes* was well supported in all topologies. In BI trees, the phylogenetic analysis of the matrices cds_fna, cds_faa, and cds12_rrna resulted in ((*Endochironomus* + *Synendotendipes*) + (*Phaenopsectra* + *Sergentia*) + (*Polypedilum* + *Stictochironomus*)) (Figure 5A, Appendix A), while the matrices cds_rrna and cds12_fna yielded (*Stictochironomus* + ((*Endochironomus* + *Synendotendipes*) + (*Sergentia* + *Phaenopsectra*)) + *Polypedilum*) (Figure 5B and Appendix A). All ML trees recovered ((((*Endochironomus* + *Synendotendipes*) + (*Phaenopsectra* + *Sergentia*)) + *Stictochironomus*) + *Polypedilum*) (Figure 5 and Appendix A). 

Due to the lack of *Phaenopsectra* and *Synendotendipes* species, the relationships of the *Polypedilum* generic complex were unclear in a recent dated molecular phylogeny [20]. Our analyses included a wider range of samples, recovering a new insight for the phylogenetic relationships within the *Polypedilum* generic complex. According to our data, *Endochironomus* + *Synendotendipes* is sister to *Phaenopsectra* + *Sergentia* (Figure 5). Although the BI and ML topologies differed, the phylogenetic relationships of the genera *Synendotendipes*, *Endochironomus*, *Phaenopsectra*, and *Sergentia* were stably supported in all trees. 

Different topologies between BI and ML trees indicated that the phylogenetic relationships based on mitogenomes among this group were still erratic, i.e., the systematic position of *Stictochironomus*, and the trees which were inferred by the heterogeneity model (CAT + GTR) were also not well supported. Therefore, we need to await further taxonomic and phylogenomic studies with more taxon sampling and availably molecular markers, such as ultra-conserved elements and single-copy orthologous genes, which have been successfully used in other insect groups [47,48,49] to explore the evolutionary history of the *Polypedilum* generic complex. 

## 4. Conclusions

Fourteen mitogenomes of six genera within the *Polypedilum* generic complex were obtained, including six complete mitogenomes and eight linear mitogenomes. All newly sequenced mitogenomes had similar structural characters and nucleotide compositions to previously published Chironomidae data. In adding *Phaenopsectra* and *Synendotendipes*, we could also reconstruct the phylogenetic relationships among the genera within the *Polypedilum* generic complex. Our results showed that *Endochironomus* + *Synendotendipes* are sister to *Phaenopsectra* + *Sergentia*, which is a new systematic finding for Chironomidae. 

## Figures and Tables

**Figure 1 insects-14-00238-f001:**
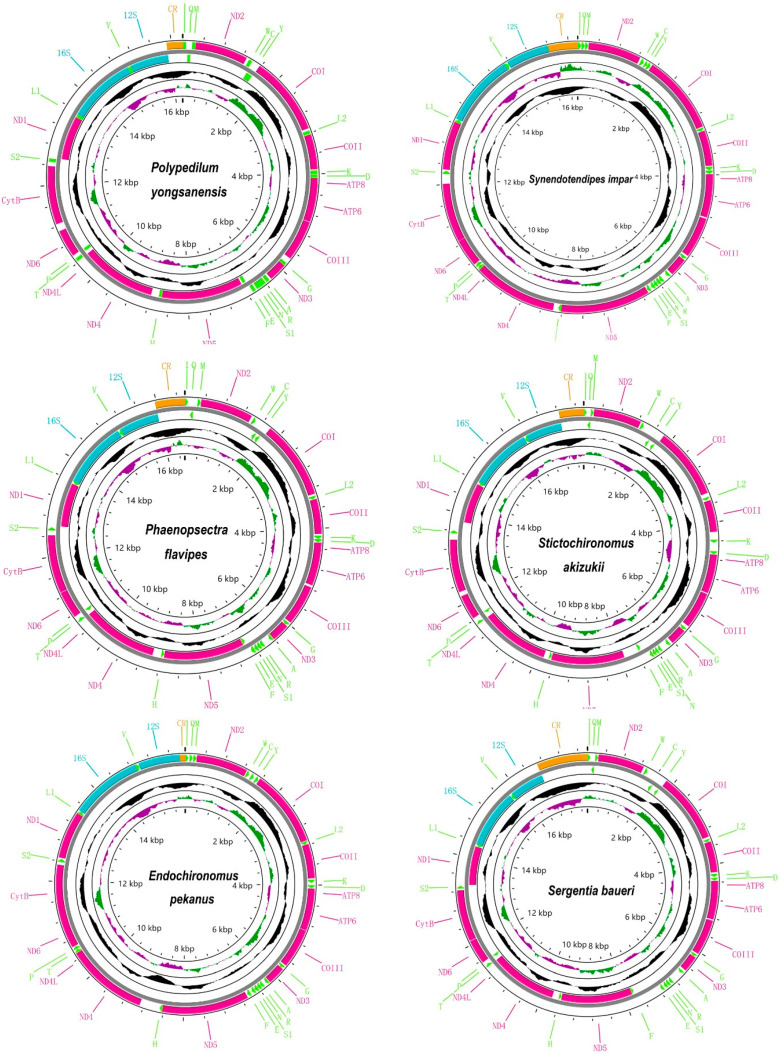
Mitogenome map showed the mitochondrial genome characteristics of representative species from six genera within the *Polypedium* generic complex. The arrow indicated the direction of gene transcription. We used normative abbreviations to represent PCGs and rRNAs, and single letter abbreviations were used to represent tRNAs. Red, green, blue, and orange represented PCGs, tRNA, rRNA, and CR, respectively. GC content of complete mitogenome showed in the second circle. GC-skew of complete mitogenome showed in the third circle. The innermost circle showed the length of complete mitogenome.

**Figure 2 insects-14-00238-f002:**
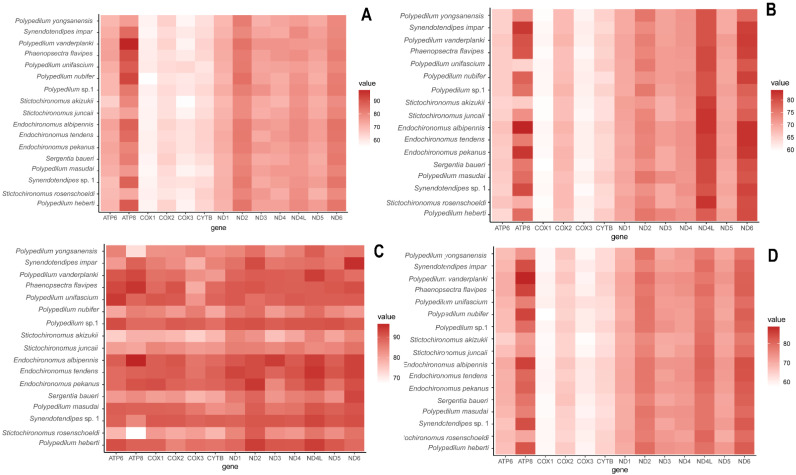
Difference in AT content of protein coding genes of *Polypedilum* generic complex mitogenomes. (**A**) First-codon positions; (**B**) second-codon positions; (**C**) third-codon positions; (**D**) first/second-codon positions.

**Figure 3 insects-14-00238-f003:**
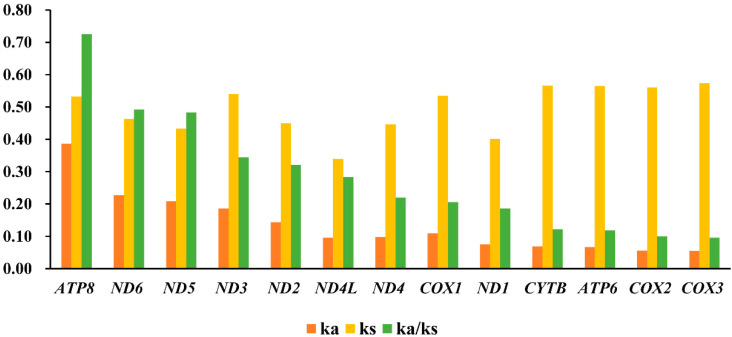
Evolution rate of 13 PCGs of the *Polypedilum* generic complex mitogenomes. Ka refers to non-synonymous nucleotide substitutions, Ks refers to synonymous nucleotide substitutions, Ka/Ks refers to the selection pressure of each PCG. The abscissa represented 13 PCGs, and the ordinate represented Ka/Ks values.

**Figure 4 insects-14-00238-f004:**
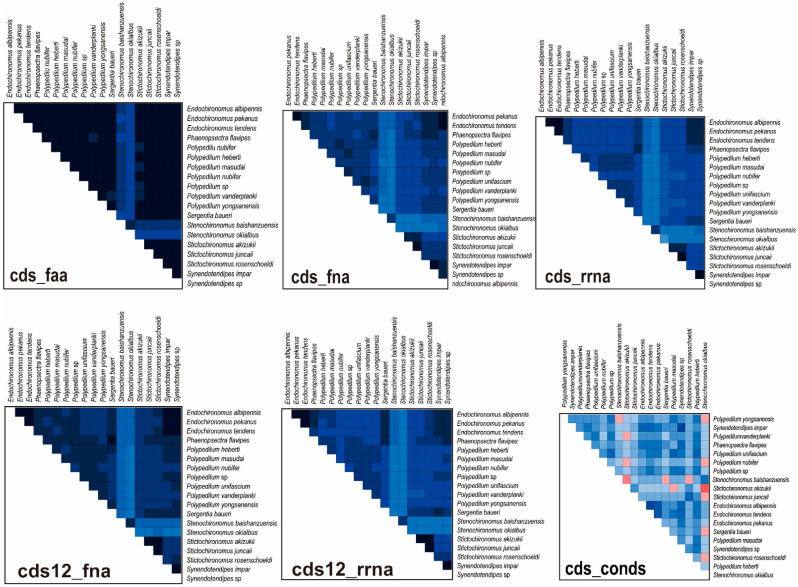
Heterogeneity analysis for different matrices. Colored squares represented pairwise Aliscore values. Score values ranged from −1 (indicated fully random similarity, dark blue) to +1 (indicated non-random similarity, bright orange, or red).

**Figure 5 insects-14-00238-f005:**
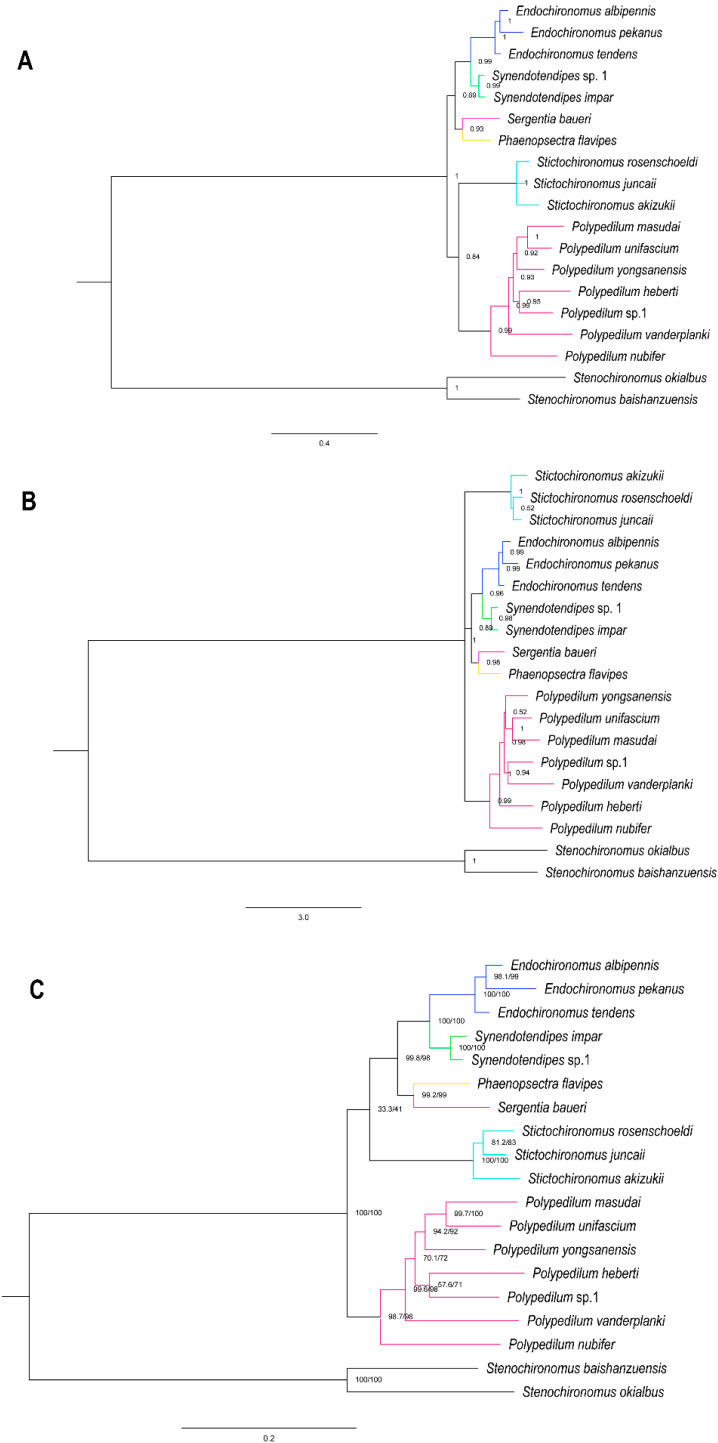
Phylogenetic trees of the *Polypedilum* generic complex: (**A**) BI tree based on analysis cds_fna in Phylobayes. (**B**) BI tree based on analysis of cds_rrna in Phylobayes. (**C**) ML tree based on the anaysis cds_faa with PMSF model in IQTREE. Support values on nodes indicate Bayesian posterior probabilities in topology A and B, while they represent SH-aLRT/UFBoot2 in topology C.

**Table 1 insects-14-00238-t001:** Collection information of newly sequenced species in this study.

Species	Location	Longitude and Latitude	Elevation (m)	Data	Collector
*Endochironomus albipennis*	Dun hua, Jilin, China	128.2360° E, 43.3247° N	511	14.VII.2016	Chao Song
*Endochironomus pekanus*	Rongjiang, Guizhou, China	108.3401° E, 26.3394° N	773	06.V.2013	Xiao-Long Lin
*Endochironomus tendens*	Dun hua, Jilin, China	128.2361° E, 43.3247° N	511	14.VII.2016	Chao Song
*Phaenopsectra flavipes*	Leigongshan Natural Reserve, Guizhou, China	108.2610° E, 26.396° N	1070	18.I.2021	Hai-Jun Yu
*Polypedilum heberti*	Gaoligongshan National Nature Reserve, Yunnan, China	98.8011° E, 25.3031° N	1536	23.V.2018	Xiao-Long Lin
*Polypedilum* sp.1	Naukluft Mountain Zebra Park, Hardap, Namibia	16.2280° E, 24.2620° N	1400	05.XII.2018	Xiao-Long Lin
*Polypedilum yongsanensis*	Tianjin Agricultural University, Tianjin, China	117.1005° E, 39.0913° N	5	16.VI.21	Xiao-Long Lin
*Polypedilum masudai*	Beihedian, Baoding, Hebei, China	115.7710° E, 39.2240° N	26	25-VII-2019	Hai-Jun Yu
*Sergentia baueri*	Qingshan Lake, Dandong, Liaoning, China	125.2560° E, 40.9939° N	196	08.VI.2016	Chao Song
*Stictochironomus akizukii*	Qingbi Stream, Cangshan, Yunnan, China	100.1430° E, 25.6475° N	2558	20.V.2018	Xiao-Long Lin
*Stictochironomus juncaii*	Wuying River, Yi chun, Heilongjiang, China	129.2470° E, 48.0869° N	283	26.VI.2016	Chao Song
*Stictochironomus rosenschoeldi*	Lian lake, Trondheim, Norway	10.3176° E, 63.3998° N	227	23.V.2016	Xiao-Long Lin
*Synendotendipes impar*	Friuli Venezia Giulia, Veneto, Italy	11.6265° E, 46.5408° N	1851	10.IX.2019	Xiao-Long Lin
*Synendotendipes* sp.1	Nima Wetland, Naqu, Xizang, China	92.0582° E, 31.7128° N	4628	05.IX.2020	Yu Peng

**Table 2 insects-14-00238-t002:** Nucleotide composition of 19 mitogenomes.

**Species**	**Whole Genome**					**PCG**					**tRNA**				
	**Length**	**AT%**	**AT-**	**GC%**	**GC-**	**Length**	**AT%**	**AT-**	**GC%**	**GC-**	**Length**	**AT%**	**AT-**	**GC%**	**GC-**
	**(bp)**		**Skew**		**Skew**	**(bp)**		**Skew**		**Skew**	**(bp)**		**Skew**		**Skew**
*Endochironomus albipennis*	15,916	79.80	0.028	20.20	−19.81	11,229	76.93	−0.170	23.07	−0.23	1497	81.56	0.030	18.44	14.49
*Endochironomus tendens*	15,790	79.42	0.025	20.58	−17.85	11,232	76.83	−0.170	23.17	0.31	1490	80.94	0.030	19.06	16.20
*Endochironomus pekanus*	16,460	79.71	0.022	20.29	−18.66	11,241	76.45	−0.180	23.55	0.04	1484	81.06	0.030	18.94	15.30
*Phaenopsectra flavipes*	16,558	79.94	0.035	20.06	−20.69	11,232	76.51	−0.180	23.49	−2.05	1500	81.60	0.030	18.40	15.22
*Polypedilum heberti*	16,506	79.75	0.007	20.17	−21.68	11,217	76.31	−0.190	23.69	−1.92	1489	81.53	0.030	18.47	12.73
*Polypedilum* sp.1	16,421	79.28	0.008	20.72	−21.81	11,223	76.28	−0.190	23.72	−3.23	1499	80.45	0.020	19.55	13.99
*Polypedilum nubifer*	15,896	77.00	0.057	23.00	−25.71	11,217	73.59	−0.182	26.41	−5.06	1488	81.12	0.017	18.88	16.01
*Polypedilum vanderplanki*	16,445	79.72	0.027	20.27	−24.96	11,214	76.40	−0.175	23.60	−2.76	1498	82.04	0.029	17.96	13.01
*Polypedilum yongsanensis*	16,226	77.01	0.050	22.99	−23.97	11,232	73.86	−0.200	26.14	−3.47	1510	80.00	0.030	20.00	14.57
*Polypedilum unifascium*	16,456	79.28	0.020	20.27	−23.68	11,196	75.91	−0.206	24.09	−2.11	1510	81.39	−0.007	18.61	13.88
*Polypedilum masudai*	15,582	78.00	−0.003	21.98	−23.68	11,223	75.28	−0.210	24.72	−2.45	1496	80.55	0.020	19.45	13.40
*Sergentia baueri*	17,810	78.27	0.040	21.71	−22.68	11,235	73.73	−0.190	26.27	−1.86	1506	80.61	0.040	19.39	12.33
*Stictochironomus akizukii*	17,773	75.98	0.075	24.02	−32.02	11,217	71.53	−0.180	28.47	−5.36	1500	79.20	0.040	20.80	11.54
*Stictochironomus juncaii*	16,850	77.16	0.064	22.84	−26.98	11,226	73.61	−0.190	26.39	−3.75	1488	79.17	0.030	20.83	16.13
*Stictochironomus rosenschoeldi*	16,713	76.92	0.055	23.08	−28.03	11,226	73.15	−0.190	26.85	−3.45	1489	79.25	0.030	20.75	13.92
*Synendotendipes impar*	16,184	78.23	0.024	21.77	−20.86	11,226	74.75	−0.190	25.25	−2.22	1490	80.74	0.040	19.26	15.68
*Synendotendipes* sp.1	16,097	79.19	0.012	20.81	−16.30	11,226	76.25	−0.180	23.75	−0.53	1500	80.80	0.040	19.20	17.36
*Stenochironomus baishanzuensis*	16,470	82.70	0.007	17.30	−21.17	11,181	79.50	−0.179	20.50	1.92	1517	84.38	0.064	15.62	14.77
*Stenochironomus okialbus*	17,893	82.21	0.014	17.79	−22.86	11,163	77.50	−0.186	22.50	−2.39	1511	84.32	0.022	15.68	17.30
**Species**	**rRNA**					**CR**					**GenBank Accession**			**Reference**	
	**Length**	**AT%**	**AT-Skew**	**GC%**	**GC-Skew**	**Length**	**AT%**	**AT-Skew**	**GC%**	**GC-Skew**					
	**(bp)**					**(bp)**									
*Endochironomus albipennis*	2252	86.01	−0.010	13.99	33.97	110	98.18	−0.1	1.82	100.00	OP950227			This study	
*Endochironomus tendens*	2246	85.89	−0.010	14.11	32.49	269	95.91	−0.1	4.09	−45.45	OP950219			This study	
*Endochironomus pekanus*	2266	85.75	−0.020	14.25	28.79	476	94.96	−0.1	5.04	−16.67	OP950228			This study	
*Phaenopsectra flavipes*	2281	85.58	−0.050	14.42	34.95	585	95.73	−0.1	4.27	−28.00	OP950216			This study	
*Polypedilum heberti*	2245	86.19	0.030	13.81	34.19	1180	92.88	−0.03	6.1	−80.56	OP950225			This study	
*Polypedilum* sp.1	2300	85.39	0.030	14.61	35.71	666	94.59	−0.1	5.41	−22.22	OP950217			This study	
*Polypedilum nubifer*	1493	85.05	−0.020	14.95	33.88	453	92.97	−0.04	7.28	−3.03	MZ747090			[23]	
*Polypedilum vanderplanki*	1491	84.95	0.000	15.05	37.29	517	95.36	−0.03	4.45	−56.52	KT251040			[24]	
*Polypedilum yongsanensis*	2288	84.88	−0.020	15.12	35.26	312	95.51	−0.1	4.49	−42.86	OP950222			This study	
*Polypedilum unifascium*	1461	84.89	0.020	15.11	33.03	627	94.42	−0.03	5.58	−65.71	MW677959			[25]	
*Polypedilum masudai*	2263	85.86	0.050	14.14	35	160	92.5	−0.20	5.62	−100	OK513041			This study	
*Sergentia baueri*	2253	85.04	−0.040	14.96	30.56	1126	92.01	−0.1	7.99	−51.11	OP950220			This study	
*Stictochironomus akizukii*	2343	84.46	−0.040	15.54	38.46	534	91.95	0.03	8.05	−58.14	OP950218			This study	
*Stictochironomus juncaii*	2308	84.32	−0.050	15.68	38.12	275	92.73	0.17	7.27	−30	OP950226			This study	
*Stictochironomus rosenschoeldi*	2319	84.73	−0.040	15.27	38.98	238	92.02	0.05	7.56	−22.22	OP950224			This study	
*Synendotendipes impar*	2224	85.07	−0.010	14.93	32.53	476	95.59	−0.1	4.41	−14.29	OP950223			This study	
*Synendotendipes* sp.1	2231	84.94	−0.030	15.06	34.52	580	96.55	−0.1	3.45	−10	OP950221			This study	
*Stenochironomus baishanzuensis*	1644	43.51	0.020	11.38	37.34	213	98.12	−0.11	1.88	0	OL742441			[21]	
*Stenochironomus okialbus*	1688	88.06	0.030	11.94	37.74	475	96.21	−0.17	3.79	33.33	OL753645			[21]	

## Data Availability

The following information was supplied regarding the availability of DNA sequences: The new mitogenomes are deposited in GenBank of NCBI and the accession numbers are OP950216–OP950228, OK513041.

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
