# Peer review of "New Mitogenomes of the Polypedilum Generic Complex (Diptera: Chironomidae): Characterization and Phylogenetic Implications"

_insects, 2023, doi:10.3390/insects14030238_

Round 1

Reviewer 1 Report

Well done!  Some minor corrections in the English are necessary; the following I have detected: line 23 'features'; line 34 'genera of the Polypedilum complex'; line 52 'sensitively perceive environmental changes in trophic'; line 65 'mitogenes in the Polypedilum generic complex'; line 70 'To reconstruct the '; line 72 'Endochironomus species'; line 78 ' six members of the Polypedilum generic group, Endochironomus...'; line 80 'were immersed in' ; line 181 'is usually used'; line 222 'within the'; line 241/242  'reconstruct'

Author Response

Thank you for reviewing our manuscript. We have addressed all comments and made significant improvements, especially in terms of language and figures. Please see our point-by-point replies in response letter.

Reviewer 2 Report

The authors have made a valuable contribution to science by producing a well written paper documenting their study of the phylogeny of the Polypedilum generic complex, a group found in a range of lentic and lotic aquatic as well as terrestrial ecosystems. The results of this study of the mitogenomes of the Polypedilum group will enable further expansion of the knowledge of Chironomidae evolutionary biology.

The tables included are clear and informative, but the figures are low resolution making the text difficult to read, unlike the high-quality images included in the Supplementary material which are legible and effective. This suggests that the poor resolution may simply be a result of reducing the image file size to enable embedding in the document.

Review comments/suggestions

Simple summary

Page 2, Lines 19-20.

This genus complex, as one of the most species rich groups has been persistently contentious.

Page 2, Line 23

….we analyzed features of the mitogenomes….

Page 2, Line 26

…….vital foundation for further study on of the……

Introduction

Page 2, Line 44

……studies on of phylogeny, evolutionary

Page 3, Line 52

Most species are sensitively and respond to percept the environmental change………

Results and Discussion

Page 5, Figure 1.  

The gene maps are low resolution making interpretation difficult because the text is eligible. The poor resolution may simply be a result of reducing the image file size to enable embedding in the document. The figures in the Supplementary material are of a much better quality. I also suggest making the arrows on the maps more obvious.

Page 8, Figure 2.  

As for Figure 1. – the resolution is low making interpretation difficult because the text is eligible.

Page 10, Figure 4. Heterogeneity analysis for different matrices – the resolution is low.

Page 11, Figure 5. Phylogenetic trees – again the resolution is low.

Conclusions

Page 12, Line 241-242

also reconstructuct reconstruct the…………

Author Response

 Thank you very much for the constructive comments. We have gone through below and made every effort to address all comments; we especially appreciate your efforts to make sure that we have provided and clarified all necessary steps in our research. Please see our point-by-point replies in  response letter. 

Reviewer 3 Report

The manuscript provides important new data on phylogenetics and systematics of Chironomidae.

In the Introduction there are few remarks that need to be attended to. The presented research was realized using standard methods and reports the results in a clear way. The quality of images/figures need to be improved for publication as they are too blurry to read in this state. Conclusion is clear and coherent. Please, re-check the formatting of the references.

Author Response

Thank you very much for the constructive comments. We have gone through below and made every effort to address all comments; we especially appreciate your efforts to make sure that we have provided and clarified all necessary steps in our research. We have re-checked the reference in the new main text. We also replaced all figures to improve them.
